# The Effect of Phase Angle on the Thermo-Mechanical Fatigue Life of a Titanium Metal Matrix Composite

**DOI:** 10.3390/ma12060953

**Published:** 2019-03-22

**Authors:** Ashley Dyer, Jonathan Jones, Richard Cutts, Mark Whittaker

**Affiliations:** 1Institute of Structural Materials, College of Engineering, Swansea University Bay Campus, Fabian Way, Skewen, Swansea SA1 8EN, UK; ashdyer2028@gmail.com (A.D.); jonathan.p.jones@swansea.ac.uk (J.J.); 2Rolls-Royce plc, P.O. Box 31, Derby DE24 8BJ, UK; richard.cutts@rolls-royce.com

**Keywords:** titanium, silicon carbide, TMF, texture

## Abstract

The thermo-mechanical fatigue (TMF) behaviour of a Ti-6Al-4V matrix composite reinforced with SCS-6 silicon carbide fibres (140 μm longitudinal fibres, laid up hexagonally) has been investigated. In-phase and out-of-phase TMF cycles were utilized, cycling between 80–300 °C, with varying maximum stress. The microstructure and fracture surfaces were studied using electron backscatter diffraction (EBSD), energy-dispersive X-ray spectroscopy (EDS), scanning electron microscopy (SEM), profilometry, and optical microscopy. The results have shown the damaging effect of out-of-phase cycling with crack initiation occurring earlier than in in-phase tests and crack propagation rates being accelerated in out-of-phase cycles. Fatigue crack initiation has been shown to be sensitive to crystallographic texture in the cladding material and thermo-mechanical fatigue test results can be considered according to a previously proposed conceptual framework for the interpretation of metal matrix composite fatigue.

## 1. Introduction

The aerospace industry is at the forefront of research, innovation, and technology. In a world where the impact of emissions on the environment is becoming increasingly understood, the challenge of efficiency is of prime concern to the industry. This is evident from the partnership between Rolls-Royce plc and the advisory council for aeronautics research (ACARE) [1]. The goals set by ACARE in their flightpath 2050 vision focus on reducing CO_2_ emissions, NO_X_ emissions, and noise pollution. This has placed huge pressure on aero-engine manufacturers to create the next generation of more efficient gas turbines. The research and development of novel materials is crucial if these goals are to be met. If the gas turbine of the future is to achieve these targets, higher overall pressure ratios, higher compressor discharge and higher turbine entry temperatures (TET) are required. An increase in TET leads to a decrease in specific fuel consumption (SFC), as fuel can burn closer to its stoichiometric value and, therefore, more efficiently. Improvements in efficiency can also be brought about through the weight reduction of components. The demand for more efficient gas turbines means materials have to be lighter than ever before, whilst at the same time they have to operate in much higher temperature environments and at higher stresses than ever before.

Materials that can help deliver a step change in performance by offering radically improved specific strength and stiffness are highly desirable. Over the last 40 years, significant improvements have been made in the design and manufacture of titanium metal matrix composites (TiMMCs). Reinforcing monolithic titanium components with silicon carbide (SiC) particles, whiskers or continuous fibres leads to a component with the high strength and stiffness of ceramics but with the damage tolerance of titanium. TiMMCs reinforced with continuous fibres are of particular interest to the aerospace industry, as the material properties are highly unidirectional. Unidirectionally reinforced TiMMCs are desirable for components that are predominantly loaded along one axis. For example, the bling and blisc application is highly promising as sizeable weight savings are predicted at the same time as improving the component’s strength and stiffness in the hoop direction.

TiMMCs have been kept out of the critical components of the gas turbine due to complex manufacturing issues. One such issue is the build-up of residual stress in the component from high HIPing (hot isostatic pressing) temperatures. This originates from differing values of the coefficient of thermal expansion for the SiC fibres and Ti-6Al-4V. Other issues are due to fibre layup and the sheer quantity of diffusion bonds present in the material leading to potential bond defects. How these residual stresses react to the complex thermal and mechanical loading conditions present in the gas turbine is of great interest [2].

The flight cycle of a jet engine consists of take-off, cruise, and landing. As this cycle is repeated day after day, components in the gas turbine are subjected to individually varying thermal and mechanical loading conditions. Thermo-mechanical fatigue testing is the most accurate way, outside of full-scale testing, to replicate the effects of thermo-mechanical loading on a test specimen. TMF tests differ significantly from isothermal tests as both thermal and mechanical loads are controlled individually depending on a desired phasing, deemed the phase angle. The most common phase angles investigated are in-phase cycles (IP, 0°), where the peak load coincides with the peak temperature, and out-of-phase (OP, 180°) cycles, where the peak load coincides with the minimum temperature. Whilst TMF testing is still a relatively novel testing technique, the standardisation of TMF has led to a much more coherent body of research with results that are more comparable and accurate. The main reason for standardisation is the improved inter-laboratory reproducibility of TMF data and with that, an increased understanding of the damage mechanisms at work. The publication of ASTM E2368-10 [3], the ISO12111:2012 [4] and the European code-of-practice for strain-controlled thermo-mechanical fatigue testing [5] have outlined testing practices and clarified testing techniques. For specimens with abnormal geometries or where using an extensometer is impractical force controlled TMF standards are also being produced [6]. Past research into TiMMCs has shown TMF loading conditions to be considerably more damaging than isothermal loading conditions [7,8]. Furthermore, phase angle effects often depend on the load applied [7]. It is clear that isothermal testing alone is insufficient for characterizing the mechanical properties of TiMMCs for realistic gas turbine applications. Evidently, further TMF testing of TiMMCs is required to contribute to the understanding of the fundamental behaviour mechanisms of the material so that future life prediction methodologies can be improved.

## 2. Experimental Methods

A Mayes ESM 100 electromechanical screw testing machine capable of loads of ±100 kN was utilised for the test programme, due to the stability of the load response under relatively slow ramp rates. The specimen alignment has received verification according to ASTM E1012-2012 [9]. The load cell has been calibrated according to BS EN ISO 7500-1:2015 [10]. A bespoke 12 kW Severn Thermal Solutions quartz lamp furnace was used to heat the specimen. The furnace uses 12 vertically mounted quartz bulbs in order to heat the test specimen and comprises of two longitudinally divided half cylinders hinged at the centre, with each half containing six lamps. Behind each lamp is a parabolic reflector, which focuses the radiant light towards the centre of the furnace. In addition, there are two quartz glass liners positioned to protect the lamps and at the same time providing a smooth channel for airflow around the furnace. Strain values were measured using an MTS632.59C-01 extensometer with a gauge length of 25 mm.

Thermal profiling to ensure compliance with the European code of practice was conducted by spot welding 6 N-type 0.25 mm diameter thermocouples (TCs) along the specimen and controlling the temperature using a thermocouple spot-welded to the centre gauge of the specimen. The thermocouple locations can be seen in Figure 1. The profiling was also critical in ensuring that a reliable and repeatable relationship existed between the shoulder of the specimen and the centre gauge of the specimen.

In order to provide effective cooling in the cycle, a modified cooling setup was designed to enhance the rate, uniformity, and repeatability of cooling. The setup consists of four quartz cooling channels connected by a central ring. The channels run parallel to the specimen, each having five air outlets directed at the specimen. These channels replaced the existing top down cooling with a more uniform distribution of cold air, leading to a more consistent thermal gradient and increased cooling rate.

As suggested by the Code of Practice [5] a control thermocouple was placed at the shoulder of the test specimen. In order to establish a reliable relationship between the shoulder and the gauge of the specimen, dynamic thermal cycles were run on a monolithic test piece. These test pieces were made purposely for thermal trials. The mismatch of coefficients of thermal expansion between the fibre and the matrix materials of the TiMMC specimens means that thermal cycles can have a damaging effect on the test piece.

During testing, thermal cycles were run between 80 and 300 °C so a thermal profile of temperatures across the gauge must be created for these bounding conditions. A profile must be established so that no thermocouples are spot welded to the gauge of the specimen during testing. The spot welding of thermocouples inside the gauge is forbidden by the code of practice [5]. Unfortunately, due to the geometry of the specimen, the relation between the specimen gauge length and the shoulder was not satisfactory for control purposes. Tie on thermocouples were originally trialled as a solution but test to test variation was observed. In order to circumvent these issues a mounted thermocouple assembly was developed. The mounted non-contact thermocouple (NCTC) consists of a ceramic tube mounted inside the furnace between the extensometer arms, shown in Figure 2. The tube holds a mineral insulated TC 1 mm from the surface of the specimen in order to avoid reduced specimen lives due to fretting fatigue. The fixed nature of the assembly means that the TC does not move from test to test, improving repeatability, another advantage over the tie-on beaded TC.

During thermal profiling this close proximity thermocouple showed a much closer relationship to the control thermocouple when compared to the tie-on thermocouple used previously. The mounted thermocouple was chosen to use for the testing phase. The relation between the thermocouples can be seen in Figure 3.

With the appropriate relationship between the NCTC and the centre of the gauge length established, overall thermal profiling of the gauge length could be undertaken. The developmental code of practice states that the axial temperature gradients within the gauge length should not exceed ±10 °C or ±2% of the temperature range [5]. In the current work, the specimen design had been focused on an appropriate specimen for the TiMMC material, rather than adhering to the TMF Code of Practice. With a non-standard specimen with an extremely long gauge length, controlling axial gradients was therefore extremely challenging and compromises were required. The axial temperature gradients were maintained according to the TMF standard across the arms of the extensometer, although not to the full extent of the parallel length of the specimen.

## 3. Material

Figure 4a shows the microstructure of the Ti-6Al-4V within the fibre region and Figure 4b shows the microstructure of the Ti-6Al-4V cladding around the composite mid-section. The microstructure of the cladding is a mixture of primary and Widmanstatten α+β phases. The microstructure within the composite region consists of primary α with intergranular β. This is shown in the lighter regions in Figure 4 and by the Electron Backscatter Diffraction (EBSD) phase map in Figure 5. After hot isostatic pressing (HIP) the microstructure within the fibre region has undergone full dynamic recrystallization and differs significantly to the crystallographic texture seen in the cladding material, where a strong basal texture is clear, Figure 6. The Ti-6Al-4V matrix material has a Young’s modulus of E = 115 GPa, a yield strength of 950 MPa and an Ultimate Tensile Stress (UTS) of 1040 MPa. The chemical composition of the Ti-6Al-4V matrix is as follows 90% Ti, 6% Al, 4% V, <0.10% C, <0.20% O, <0.05% N, <0.0125% H and <0.3% Fe by weight.

An important aspect of the microstructure is the size of the recrystallized grains within the 50 µm deposited layer. This is highly dependent on the thermo-mechanical conditions imposed during processing. SEM and EBSD examinations show that the dynamically recrystallised microstructure within the fibre region is non-homogeneous, Figure 6. EBSD maps show a relatively finer grain size surrounding the fibres when compared to the outer region of the 50 µm deposited layer. This is due to a higher level of stress, due to differing values of the coefficient of thermal expansion (CTE), around the SiC fibres leading to a more noticeable extent of dynamic recrystallization [11].

The fibres used are SCS-6 fibres, which are 140 μm longitudinal fibres, laid up hexagonally. At the centre of the fibres is a carbon core. Chemical vapour deposition (CVD) is used to coat the fibre with a 1.5 μm layer of pyrolytic carbon, a 15 μm non-uniform composition SiC coating, a 35 μm coating of stoichiometric SiC and a 3 μm outer carbon coating [12]. The carbon coating exists to stop the reaction between SiC and titanium as well as creating a weak interface to aid de-bonding. The SCS-6 fibre is then coated in 50 μm of titanium using electron beam physical vapour deposition (EBPVD). The fibres are laid up and subjected to a two stage HIP, with a peak temperature and pressure of 925 °C and 100 MPa respectively, in order to consolidate the fibres. The test matrix was chosen in order to investigate the effect of the most extreme phase angles on the TiMMC specimens. Tests were undertaken under IP 0° conditions as well as OP 180°. The temperature range chosen was 80 to 300 °C. As TiMMCs are envisioned for compressor applications a maximum temperature (Tmax) of 300 °C is a good representation of service conditions. Ideally the minimum temperature would be room temperature to match in service flight cycles. However, cooling the specimen down to room temperature requires a vast amount of cooling leading to a large increase in cycle time. A minimum temperature (Tmin) of 80 °C was chosen as a compromise between representative conditions and achievable experimental cycles.

## 4. Results

The test conditions for the programme are shown in Table 1, and the results can be seen in Figure 7. The R value was kept constant at R = 0.1, using an 80-s cycle (f = 0.0125 Hz) and the maximum stress was varied in order to investigate the influence of σ_max_ on TMF behaviour.

Failures within the tested specimens differed significantly dependent on phase angle. For out-of-phase (OP) tests, failures were generally within the gauge length whereas for in-phase (IP) tests failures often occurred at the shoulder of the specimen. This proved problematic due to the length of the specimens, since only the gauge length between the extensometer arms is controlled within the limits prescribed by the TMF stress controlled code of practice [6]. Outside of this region the temperature profile decays and the extremes of the parallel length are at a lower temperature than the centre of the gauge length. In most materials, this does not pose a problem since the higher temperatures at the centre promote earlier failures. However, in materials such as titanium metal matrix composites, fatigue lives often show a nonlinear relationship with temperature due to the relaxation of residual stress. As such, many of the failures in Figure 7 (particularly for IP tests) should be considered a minimum fatigue life (and as such are marked with arrows) since the material in the centre of the gauge length experiencing the controlled temperature conditions has not failed within this number of cycles.

Although the failure locations prove problematic, clear trends can be seen in the data with IP lives greatly exceeding OP fatigue lives. It is also possible to subdivide the results into three distinct regions, which can be discussed in terms of the conceptual framework for interpretation of metal matrix composite (MMC) fatigue discussed by Talreja [13]. Failure in region I, the static region, of the fatigue life diagram is primarily due to fibre failure and the catastrophic ductile failure of the matrix. The location of region I depends on the failure strain of the fibres. Region II, the progressive region, is dominated by de-bonding of the fibres, matrix cracking and fibre bridging. Finally, region III is the fatigue limit region. Region III is defined by the fatigue limit of the matrix material. Cracks may initiate in region III however, they remain confined between fibres as fibre bridging works to arrest crack growth.

It is evident from Figure 7 that OP TMF tests exhibit reduced lives when compared to IP loading under 1200 MPa. The figure suggests that this is when IP loading conditions enter region I. The transition from matrix dominated failure into fibre dominated failure is evident through the large drop in fatigue life as well as the ductile appearance of the fracture surface, seen in Figure 8.

Region II is visible for both IP and OP tests. For these conditions, matrix dominated failures occur widely. Typically, the OP failures initiate from subsurface quasi cleavage facets of the type described by the Evans–Bache model [14], and widely seen at high stresses in Ti6Al4V [15]. With an increase in stress comes a reduction in crack area upon failure. It can be seen in Figure 9 that the area encompassed by the crack reduces with higher stress and with OP conditions. This is consistent with Griffith’s theory that strength is inversely proportional to the square root of the crack length [16]. In IP specimens where failures were generally outside of the temperature-controlled region, the intermediate temperature experienced as the temperature profile decayed meant that residual stresses were not a significant influence on the fatigue life. Furthermore, since the formation of subsurface facets in the Evans–Bache model diminishes at higher temperatures, as critical resolved shear stress values for basal and prismatic slip show less disparity, this mechanism for crack initiation is reduced, and surface failures prevail, as shown in Figure 10.

The much larger region II for OP loading conditions is due to high stresses at minimum temperatures leading to reduced ductility. At low temperatures, axial residual stresses in the matrix are added to the applied stress, the result is more matrix dominated failure. IP results show a smaller region II and a higher fatigue limit. This is due to creep at high temperatures promoting a cyclic stress relaxation of the matrix, leading to a higher fatigue limit. However, at high temperatures the creep of matrix material adds an additional load onto the fibres leading to fibre dominated failures. This could explain how region I occurs at a lower applied stress in IP specimens.

Interestingly, the first IP test conducted at 1000 MPa shows a significantly reduced life than other IP tests. On inspection of the failed specimen (with failure in the controlled region of the gauge length), failure was found to have initiated at a large area of faceting due to a zone of macro-texture [17] in the cladding material, shown in Figure 11. EBSD was used to confirm that the faceting present more generally on the surface of this specimen was basal in orientation, Figure 12. Individual facets were found along with denser regions of basal orientation resulting in macrozone formation. A retest was conducted, and the life achieved is shown in Figure 7 to be within region II and more consistent with the other data.

In order to investigate the rate of crack growth in the test specimens and also the overall contribution to the total life of the both crack initiation and crack propagation, striation counts were undertaken on all specimens. The striation count was conducted using scanning electron microscopy with roughly 30 images captured per specimen. Five sets of ten striations were taken and an average length for one striation was calculated. The locations of the striations were plotted using the co-ordinates from the SEM stage. The striation counts were used to calculate the life to first striation for each specimen as well as calculating crack length as a function of number of cycles. Whilst it is recognised that striation-based crack growth may not indicate the true length of crack propagation, with facet-based growth also contributing, it was hoped to provide an indication of phase angle effects.

Comparing counts taken from OP and IP specimens, it was found that OP specimens spent a much smaller percentage of total life before the first striation, suggesting that cracks form much earlier in the life of OP specimens. Striation density was also found to be much less in OP specimens, meaning that cracks propagate at a much higher rate. This is understandable as the matrix is subjected to a higher maximum stress under OP conditions. Moreover, striation density on average was found to be higher within the composite region, Figure 13. Therefore, calculated lives to first striation were longer when counts within the composite region were included in the calculation. This suggests that fibre bridging leads to a reduction in the crack growth rate. Measurements taken within the composite region were also found to be much more varied than in the cladding region. This suggests that the crack growth rate is varied, this is to be expected, as the crack growth rate is known to fluctuate with fibre bridging and failure [18]. 

In measuring the pull out lengths of the fibres, Figure 14, the positive pull-out lengths were measured across the surface of each specimen where the crack had propagated through the composite region. It is apparent from the results that the fibre pull-out length decreases with an increase in applied stress. It is also apparent that OP tests make up the majority of the pull-out measurements. The propagation of cracks through the composite region is most prominent at lower fibre stresses and characteristic of matrix dominated failure. OP failure is typically matrix dominated as high stresses at low temperature lead to a higher peak stress in the matrix. OP tests also have a lower peak fibre stress allowing cracks to propagate further through the composite region due to crack bridging by intact fibres. This would explain why there are more pull-out measurements for OP tests when compared to IP conditions where failure is predominantly dominated by fibre failure.

The literature states that OP test conditions create fracture surfaces with lower average pull-out lengths than IP test conditions [19]. This is due to the residual stress state of the matrix during OP testing. After consolidation, when the composite is cooled to room temperature, there is a compressive stress at the interface between the fibre and the matrix leading to the fibre being in compression. At high temperatures, the compressive stress acting on the fibre is relaxed, this allows more de-bonding to occur in the wake of the crack tip and thus leads to longer pull-out lengths than at lower temperatures. During OP testing peak stress is applied at a minimum temperature and therefore a lower pull-out length will be recorded. This trend is seen with specimens tested at 900 MPa under both IP and OP conditions, however more low stress tests would be needed in order to fully justify this hypothesis.

The reduction in fibre pull-out length with applied stress can be explained by the higher stress at the crack tip. Higher applied stresses lead to fibre failure before de-bonding has chance to reduce the effective stress concentration at the crack tip.

## 5. Conclusions

The following conclusions can be drawn from the current work
A test facility has been developed upon which TMF tests have undertaken. Due to the difficult nature of controlling transient temperature, temperature could only be controlled within the extensometer gauge length (25 mm). Failures outside of this region therefore led to the declaration of a minimum life for these tests, most often in IP tests.Phase angle in TMF cycles has been shown to have a significant effect on the fatigue life of TiMMCs. OP test conditions have led to severely reduced fatigue lives. In OP tests, cracks were found to initiate earlier and to propagate more quickly, since the matrix is subjected to higher stresses.A change in the failure mode was seen at 1200 MPa under IP conditions. The specimen exhibited a severely reduced life, lower than the corresponding OP test, and the fracture surface showed no signs of fatigue failure. It is therefore concluded that a transition from a matrix dominated failure to a fibre dominated failure was seen.The TMF test results were found to be consistent with the conceptual framework for the interpretation of MMC fatigue proposed by Talreja [13]. In region I, the static region, of the fatigue life diagram is primarily due to fibre failure and the catastrophic ductile failure of the matrix. Region II, the progressive region, is dominated by de-bonding of the fibres, matrix cracking and fibre bridging. Finally, region III is the fatigue limit region and defined by the fatigue limit of the matrix material.Cracks were found to initiate in one of two ways. Subsurface initiations occurred due to macro zones and flat faceted regions, whereas surface cracks initiated from large grains on the surface of the specimen.

## Figures and Tables

**Figure 1 materials-12-00953-f001:**
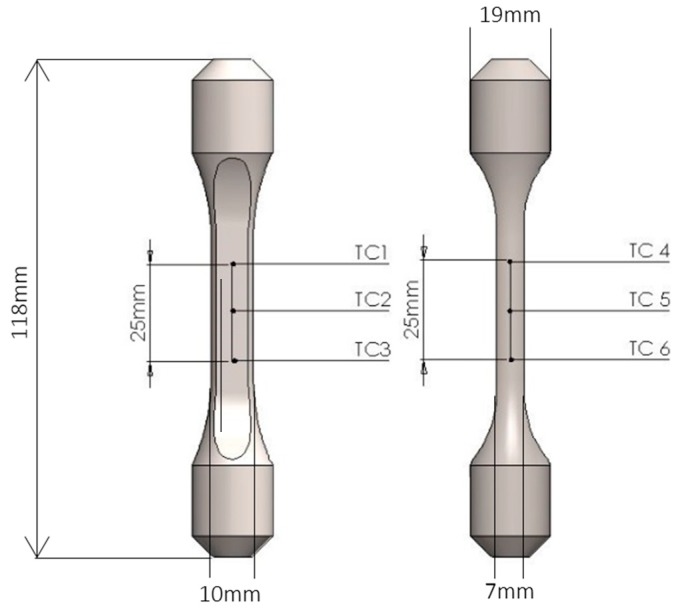
Thermocouple positions for thermal profiling. The positions of the outermost thermocouples define the position of the extensometer arms and the region of temperature-controlled material in the TMF cycle.

**Figure 2 materials-12-00953-f002:**
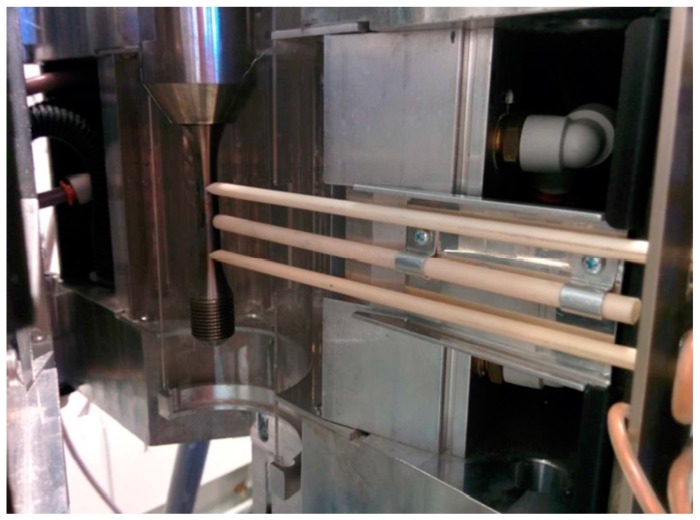
Close proximity thermocouple mount, allowing for more direct temperature readings to be made from the gauge length and hence derive a consistent relationship with the specimen surface temperature.

**Figure 3 materials-12-00953-f003:**
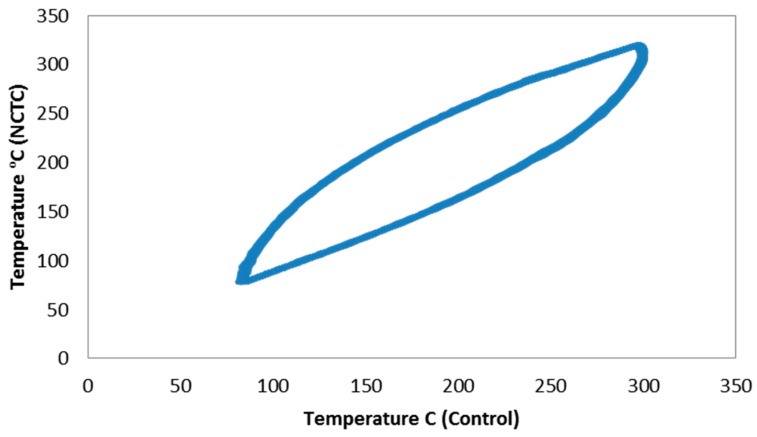
Relationship between the centre gauge thermocouple and the close proximity mounted thermocouple over 50 cycles.

**Figure 4 materials-12-00953-f004:**
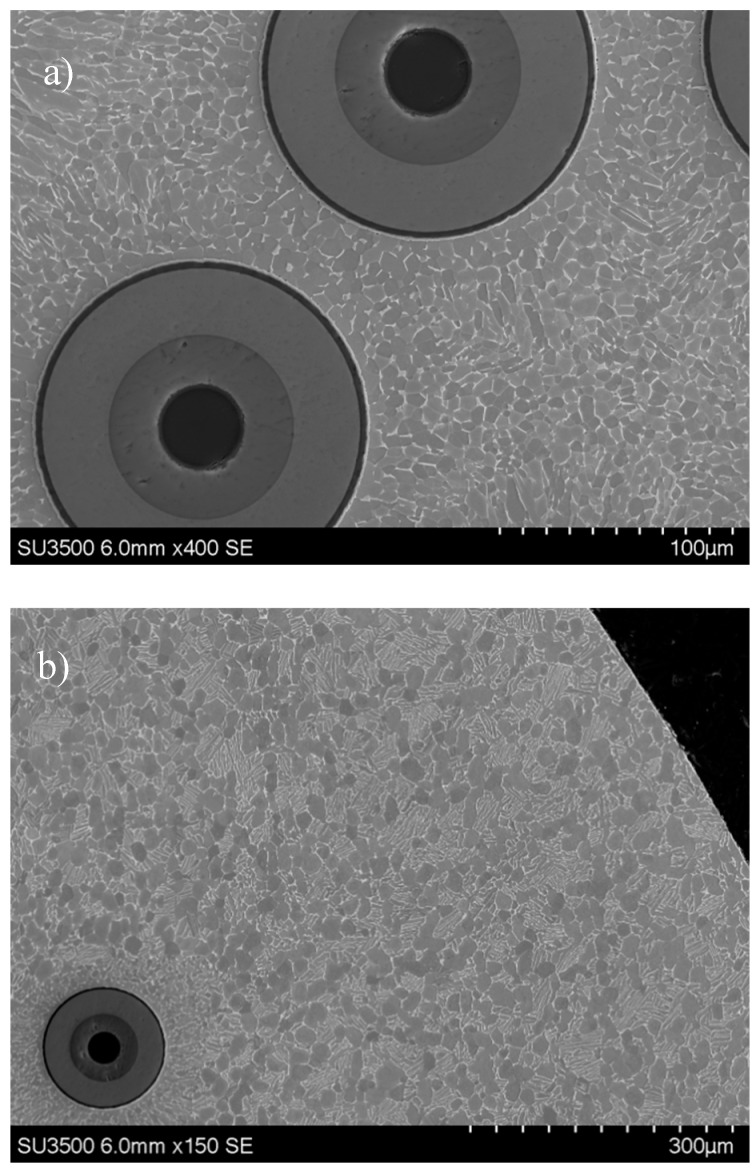
Microstructure of TiMMC material. A significant difference is evident between (**a**) the regions around the reinforcing fibres and (**b**) the cladding area on the outside of specimens.

**Figure 5 materials-12-00953-f005:**
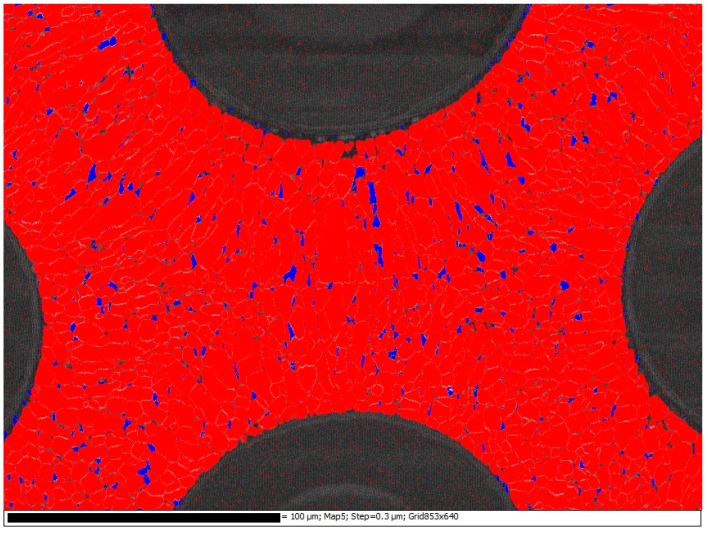
EBSD phase map showing primary α in red with intergranular β in blue.

**Figure 6 materials-12-00953-f006:**
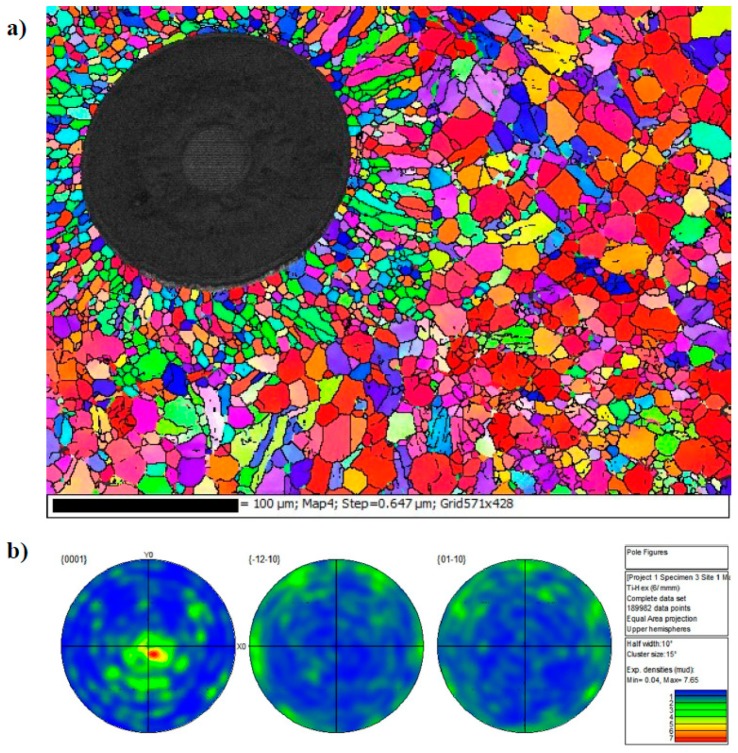
(**a**) Texture map using inverse pole figure colouring indicating grain development as a function of the HIP procedure. (**b**) Pole figures indicating strong basal texture of overall material.

**Figure 7 materials-12-00953-f007:**
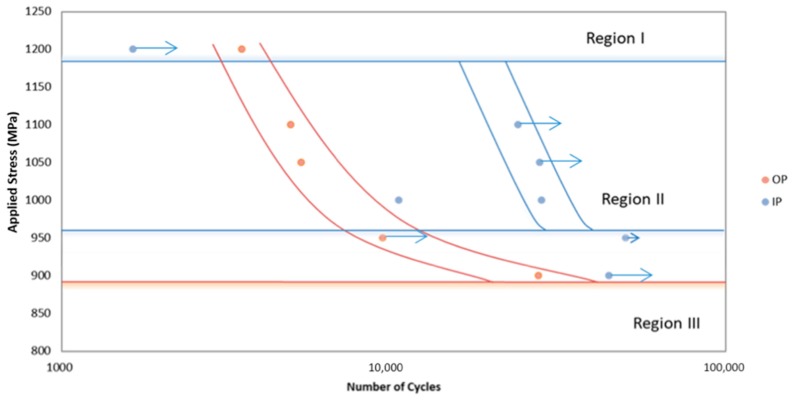
S–N curve for tests performed under both in-phase and out-of-phase loading at 80–300 °C in Ti-MMC. Short arrows indicate run out tests, longer arrows indicate tests, which failed outside of the temperature-controlled gauge length.

**Figure 8 materials-12-00953-f008:**
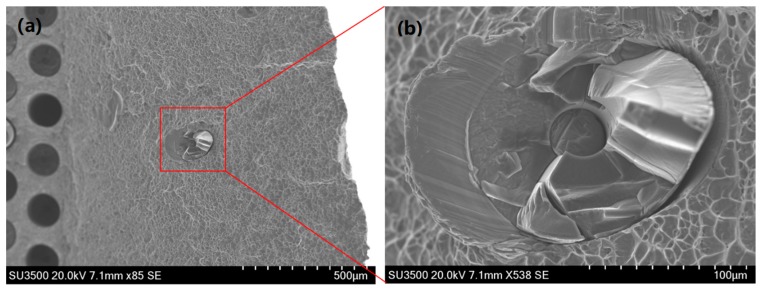
Ductile failure in out-of-phase test at 1200 MPa. (**a**) SEM image of fibre fragment within the cladding region; (**b**) higher magnification SEM image showing ductile failure and fibre contact marks.

**Figure 9 materials-12-00953-f009:**
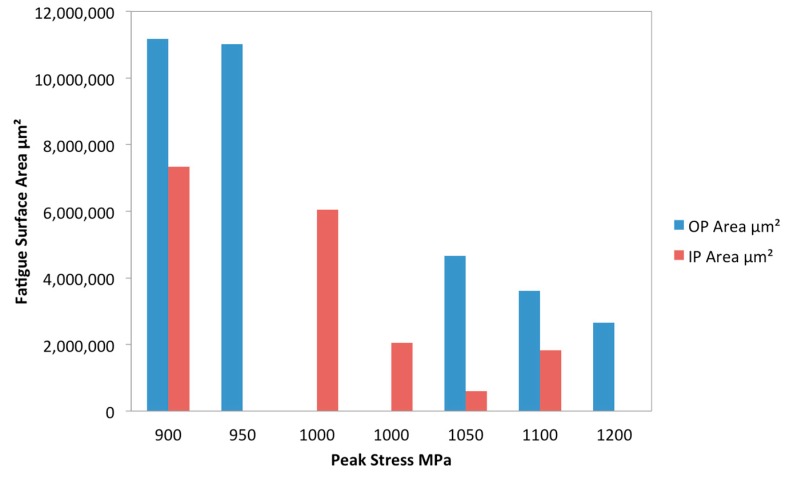
Bar chart showing the fatigue crack area against peak applied stress for all TMF tests conducted where fatigue cracks were present.

**Figure 10 materials-12-00953-f010:**
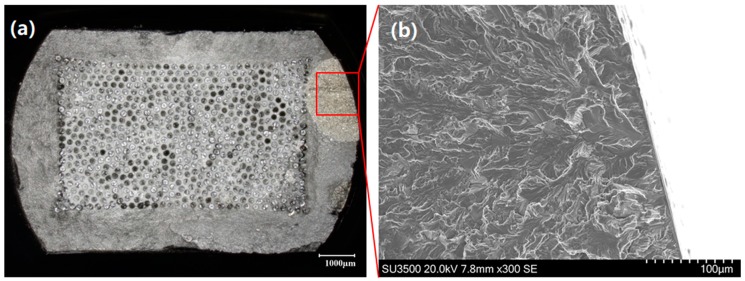
Surface crack initiation in in-phase test at 1100 MPa, where the subsequent crack grows first through the matrix material and then into the fibrous region. (**a**) Optical microscopy of the full fracture surface; (**b**) SEM image of primary surface crack initiation site.

**Figure 11 materials-12-00953-f011:**
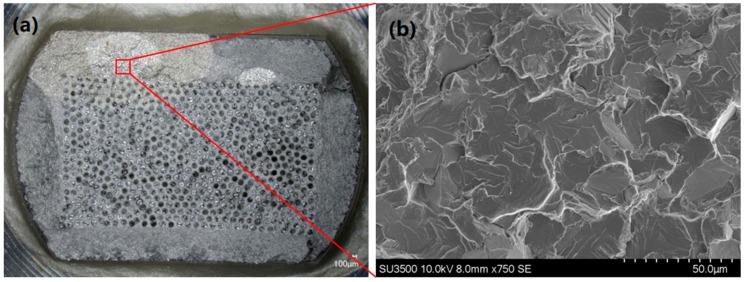
Early failure in IP 1000 MPa specimen shown to be related to a zone of macrotexture indicated by extensive facet formation. (**a**) Optical microscopy of the full fracture surface; (**b**) SEM image of flat faceted area at the crack initiation site.

**Figure 12 materials-12-00953-f012:**
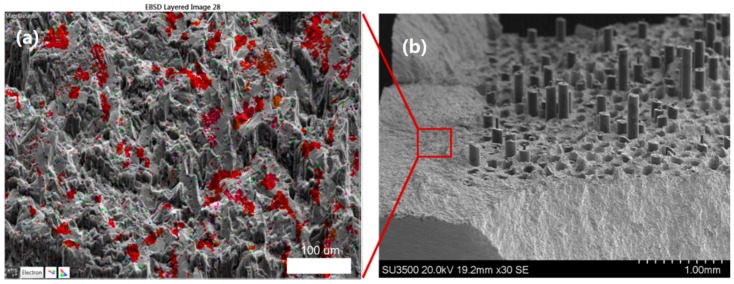
High density of basal facets discovered in IP 1000 MPa cladding area, which gave rise to crack initiation and premature failure. (**a**) IPF Z EBSD map of fracture surface imposed over tilt compensated SEM image; (**b**) low magnification SEM image of tilted stage before EBSD analysis.

**Figure 13 materials-12-00953-f013:**
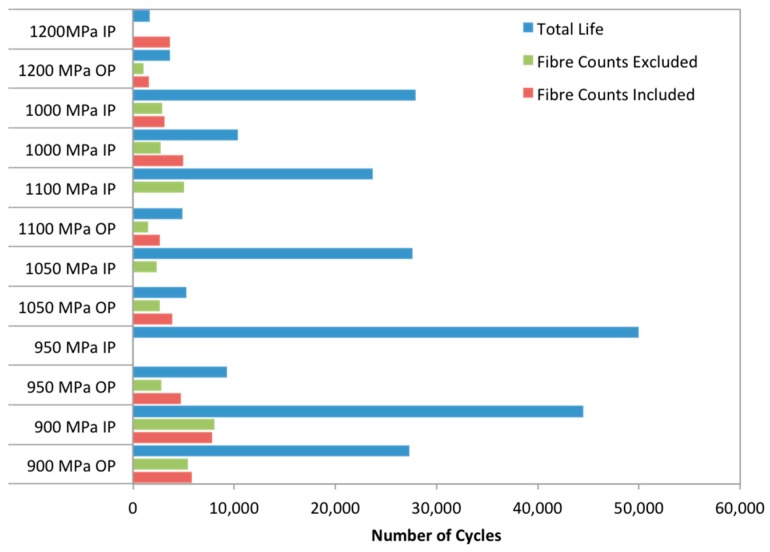
Striation counts as a proportion of total life of tested specimens.

**Figure 14 materials-12-00953-f014:**
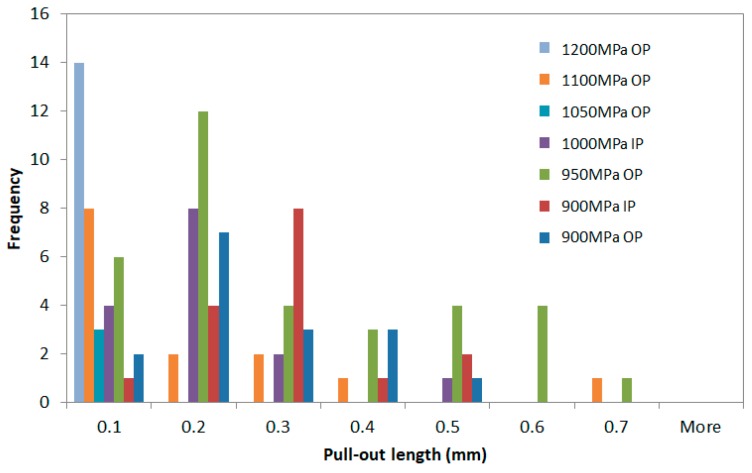
Pull out lengths of fibres for IP and OP specimens.

**Table 1 materials-12-00953-t001:** TMF test matrix for 80–300 °C temperature cycle, with test conditions shown for in-phase and out-of-phase tests.

Temperature Cycle, 80–300 °C, R = 0.1						
Phase Angle (°)	Maximum Stress (MPa)					
0° In-Phase	900	950	1000	1050	1100	1200
180° Out-of-Phase	900	950	-	1050	1100	1200

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
