# Peer review of "The Effect of Phase Angle on the Thermo-Mechanical Fatigue Life of a Titanium Metal Matrix Composite"

_materials, 2019, doi:10.3390/ma12060953_

Round 1
Reviewer 1 Report
The submitted manuscript entitled ‘The effect of phase angle on the Thermo-Mechanical Fatigue life of a Titanium Metal Matrix Composite’ deals with the complex and coupled materials investigation of a Ti matrix composite planned to be used in aircraft industry. The manuscript is interesting and worth to publish; however, in the opinion of this Reviewer, the reliability of the test method does not comply the strict conditions of aircraft industry. Besides a list of technicalities, listed below arose.
- Please solve every abbreviation at its first appearance, even if it is well-known and even if it is in the Abstract.
- In the Abstract, please give a hint about the size (diameter, length) of the SiC fibres.
- Please use subscripts in the case of CO2 and NOx.
- The dimensions of the samples are missing.
- How was the constant temperature ensured in the radial direction? Was it calculated based ont he conductivity of the sample?
‑ Please always let a space between the values and their dinómensions, except in the case of ’°C’ and ’%’.
- How were the fibres laid up?
- What was the test frequency?
Author Response
A point by point discussion is provided in the attached document. The authors are grateful to the reviewers for their helpful and instructive comments

Reviewer 2 Report
The paper presents intresting results, but it seems that in several part of the manuscript it is a strong lack in experimental data. I leave a few comments below in order to improve the quality of manuscript.
Page 1 line – 24 – CO2 (2 should be formatted)
2. Ti-6Al-4V – Please add chemical composition and basic mechanical properties
3. Fig. 4 – What is a what is b ? Please improve the marks (a) and (b)
4. Page 7 line 184 – test matrix is in Table 1 – not Fig. 7
5. Table 1 – please consider the revision of the Table caption
6. Fig. 7 – Please specify the number of tested specimens? It seems that for statistical analysis of fatigue S-N curves, the number of specimens is insufficient. Please add more results and draw confidence level for mean fatigue life
7. Fig. 7 – y-axis – should be MPa
8. Fig. 8 – An additional image with higher magnification is required
9. Fig10. Also the same comment – add error bars (but now probably it is impossible due to a lack of additional fatigue test)
10. Fig. 11 – The image is fuzzy – please improve the quality
11. Fig. 12 – Please improve it – it seems that is a part of other Figure (b) – improve scale bar
To conclude:
I have also heavy doubts about the fact that for IP specimens - all were broken far from gage length - it needs more comments and investigations. By the way - the weakest point of this paper is too small number of specimens subjected for experimental campaign.
Author Response

(The authors gave the same response as above.)

Round 2
Reviewer 1 Report
Two problems / question remained unanswered:
In the Abstract, please give a hint about the size (diameter, length) of the SiC fibres.
How were the fibres laid up?
The reason of the Authors is: '...these details are considered proprietary by the industrial partners.'.
This reason is unacceptable.
All of the experiments described in the manuscript have to be repeatable by the readers. If the Authors hold their opinion, then the manuscript is not ready for publication and cannot be published.
Author Response
The industrial partners have agreed to allow this detail in the paper for publication.
Details (140um longitudinal fibres, layed up hexagonally) have been added in the main body of the paper and in the abstract. Since SCS-6 fibres are commercially available we have not put their full diameter details in the abstract, rather they are described in detail in the body of the paper.
We hope that the reviewer finds these revisions acceptable.
Reviewer 2 Report
Accepted in presented form
Author Response
No revisions required
Round 3
Reviewer 1 Report
Thanks for all the changes and corrections.